# Displacement Sensing Using Bimodal Resonance in Over-Coupled Inductors

**DOI:** 10.3390/s25061822

**Published:** 2025-03-14

**Authors:** Alexis Hernandez Arroyo, George Overton, Anthony J. Mulholland, Robert R. Hughes

**Affiliations:** 1School of Electronic, Electrical and Mechanical Engineering, Faculty of Engineering, University of Bristol, Bristol BS8 1TR, UK; zu18921@bristol.ac.uk (G.O.); robert.hughes@bristol.ac.uk (R.R.H.); 2School of Engineering Mathematics and Technology, Faculty of Engineering, University of Bristol, Bristol BS8 1TR, UK; anthony.mulholland@bristol.ac.uk

**Keywords:** inductors, coupling coefficients, mutual inductance

## Abstract

This paper presents the theory and key experimental findings for investigating the generation of bimodal resonance (frequency-splitting) phenomena in mutually over-coupled inductive sensors and its exploitation to evaluate relative separation and angular displacement between coils. This innovative measurement technique explores the bimodal resonant phenomena observed between two coil designs—solenoid and planar coil geometries. The proposed sensors are evaluated against first-order analytical functions and finite element models, before experimentally validating the predicted phenomenon for the different sensor configurations. The simulated and experimental results show excellent agreement, and first-order best-fit functions are employed to predict displacement variables experimentally. Co-planar separation and angular displacement are shown to be experimentally predictable to within ±1 mm and ±1° using this approach. This study validates the first-order physics-based models employed and demonstrates the first proof of principle for using resonant phenomena in inductive array sensors for evaluating relative displacement between array elements.

## 1. Introduction

Precise measurement of displacement is critical in a wide range of engineering applications. These include robotic sensing and control [1] as well as for measuring both strain within rigid structures [2] and relative displacement between components [3,4,5]. There are a wide range of available techniques that can be employed, depending on the requirements of the application, employing different physical phenomena. These include optical and laser methods [6,7,8], electrical [9,10], and electromagnetic [11] principles.

A range of different electromagnetic techniques have been explored in the literature for displacement, including capacitive, inductive, and magnetic displacement mechanisms, all using electromagnetic fields to convert relative displacement into an electrical signal [11]. One of the main techniques for measuring displacement involves the use of magneto-inductive principles [12,13], which measures the electrical properties of an inductive coil as a metallic test specimen is displaced. This relies on the change in mutual inductance between the coil and eddy currents generated within the metal sample. Jiao et al. [14] explored the mutual inductance between an LC resonator and conductive material, using this mutual inductance to calculate the separation and displacement between a resonator and the material. Similar magnetic techniques have employed magnetoresistive or Hall-effect sensors [15] to quantitatively measure the magnetic field increase or decrease from a permanent or electro-magnet due to displacement. Most of the magnetic displacement devices that are commercially available employ Hall-effect sensors, and existing technologies for human–computer interfacing measure angular and lateral displacement by evaluating the magnetic field changes of permanent magnets using Hall-effect sensors [16,17].

The tracking of resonant frequencies in inductive coils has been used to measure displacement for several applications [18,19], including by Bonfitto et al. [20] who developed a radial displacement sensor for active magnetic bearing systems.

Alternative approaches to non-contact displacement measurements include the use of wireless passive LC sensors, where strongly-coupled LC resonators present a frequency-splitting phenomenon, studied by Zhang et al. [21,22] for power transfer applications. There, the maximum efficiency is typically at the resonant frequency of the resonators; however, when similar resonators are positioned in close proximity, the mutual inductance between them increases significantly, resulting in a split in the resonance frequency and a decrease in the efficiency of power transfer. However, the phenomenon of frequency-splitting is not limited solely to power transfer applications.

The frequency-splitting phenomenon has been extensively studied in microwave antenna theory, particularly in the context of analysing displacement in microwave sensors through frequency response [23,24]. The use of the frequency-splitting phenomenon is present in different radio-frequency research studies conducted by Babu and George [25,26], where they illustrated a linear and highly sensitive displacement measurement system for wireless passive LC sensors. The frequency-splitting phenomenon manifests when inductors exhibit a substantial mutual inductance, a property determined by the magnetic coupling coefficient, *k*, and depends on the amount of flux sharing between inductors. As such, the frequency-splitting phenomenon is highly dependent on the distance and alignment of inductors. It has therefore been effectively applied to measure displacements in such sensors [27,28] and used across a range of different applications. Among these applications is the measurement of fluid levels inside a tank, where a passive resonator coil floats at the liquid’s surface and the frequency-splitting effect is measured in an external coil [27]. Frequency-splitting frequency has also been employed in medical applications [29] where an optimum frequency tracking method ensures that the inductive link is working at resonance and the voltage and efficiency are at maximum.

The use of magnetic coupling for evaluating displacement is not limited to the use of two LC resonators. To date, the evaluation of displacement in inductive resonant sensors has primarily focused on coaxial air-core coils. This paper seeks to extend the principles of resonant frequency-splitting to over-coupled inductors in planar array configurations to evaluate relative displacement (separation and angular) in adjacent coils. In this paper, we exploit the magnetic coupling enhancing effects of ferrite cores and employ simple approximations to link the frequency-splitting to the relevant displacement variables via the coupling coefficient.

The proof of principle is demonstrated in a two-coil configuration, where the evaluation of the coupling coefficient is made by monitoring the frequency spectra response in the active branch of LC sensors. The wireless and single-branch measurement allows the application of the sensors in different technologies, yet to be explored. The different coil configuration extremes demonstrated in this paper highlight the accuracy and versatility of the modelling for designing resonant displacement sensors.

## 2. Theory

Inductive array sensors operate on the principle of electromagnetic induction, where alternating currents generate changing magnetic fields via Ampere’s law. These changing magnetic fields generated by a transmitter coil generate induced currents in neighbouring elements [30]. For closely spaced coils, this effect generates mutual coupling between them. This paper focuses on the prediction, measurement, and characterization of coupling between identical resonant inductors for the evaluation of displacement in inductive sensors. This exploits a phenomenon of bimodal resonance observed by Zhang et al. [25,31] and Hughes et al. [32] in closely packed inductors. The following subsections outline the principles behind mutual coupling and resonance in two-coil probes.

### 2.1. Self-Resonance Equivalent Circuit Model

Inductive sensors can be modelled as a parallel LC equivalent circuit (see Figure 1a), with the inductor, $L_{1}$, and the capacitor, $C_{1}$, representing the energy stored in the electrical field within the sensor, respectively [33]. Typically, $C_{1}$ is the lumped combination of capacitive effects between coil windings as well as any cabling or capacitive loads applied to the sensor. The resonant frequency, $f_{0}$, can be defined for a single inductor circuit as(1)$$\omega_{0}=2\pi f_{0}=\frac{1}{\sqrt{LC}},$$
where $\omega_{0}$ is the angular resonant frequency. At this natural resonance, the magnitude of the impedance, |Z|, of the sensor is a maximum, and the voltage, *V*, and current, *I*, within the circuit are in phase (zero phase lag).

Operating inductive sensors at or near resonance have been shown by many authors to improve measurement sensitivity and maximise the mutual inductance between array elements [34,35,36]; however, the resonant measurement techniques in array sensors have yet to be explored in detail.

### 2.2. Two-Coil Mutual Resonance Model

When close to another comparable coil, the primary coil will inductively couple to the secondary coil (see Figure 1a). The secondary coil can be modelled as an inductor, $L_{2}$, in series with a resistor, $R_{2}$, and a capacitor, $C_{2}$. This coupling, parametrized by the coupling coefficient, *k* (see Equation (Equation 10)), will alter the effective inductance and resistance ($L_{1}^{\prime }$ and $R_{1}^{\prime }$, respectively) of the primary measurement circuit and will distort the measured impedance, $Z_{m}^{\prime }$, given as [34](2)$$\begin{array}{cc} Z_{m}^{\prime } & =\frac{R_{1}^{\prime }+i\omega L_{1}^{\prime }}{1+i\omega R_{1}^{\prime }C_{1}-\omega^{2}L_{1}^{\prime }C_{1}}. \end{array}$$Here, $L_{1}^{\prime }$ and $R_{1}^{\prime }$ are given by(3)$$\begin{array}{cc} L_{1}^{\prime } & =L_{1}\left[1-\alpha^{2}\frac{L_{2}}{L_{1}}\left(1-\frac{\omega_{2}^{2}}{\omega^{2}}\right)\right], \end{array}$$(4)$$\begin{array}{cc} R_{1}^{\prime } & =R_{1}\left[1+\alpha^{2}\frac{R_{2}}{R_{1}}\right], \end{array}$$
where $\alpha^{2}$ is defined as(5)$$\alpha^{2}=\frac{\omega^{2}M^{2}}{R_{2}^{2}+\omega^{2}L_{2}^{2}{\left(1-\frac{\omega_{2}^{2}}{\omega^{2}}\right)}^{2}},$$
where $M=k\sqrt{L_{1}L_{2}}$ is the mutual inductance between coils. The resonant frequencies of the two-coil system can be derived from Equations (Equation 2)–(Equation 5) to give the resulting expression (see Appendix A):(6)$$\begin{array}{c} \omega_{\pm }^{\prime }=\sqrt{\frac{1}{2}\frac{\omega_{1}^{2}}{\left[1-k^{2}\right]}\left[1+\frac{\omega_{2}^{2}}{\omega_{1}^{2}}\left(1\pm \frac{\omega_{1}}{\omega_{2}}\sqrt{\frac{\omega_{2}^{2}}{\omega_{1}^{2}}+\frac{\omega_{1}^{2}}{\omega_{2}^{2}}+4k^{2}-2}\right)\right]}, \end{array}$$
where $\omega_{n}=\sqrt{1/L_{n}C_{n}}$ is the uncoupled resonant frequency of coil *n*. Equation (Equation 6) predicts two distinct resonant frequencies of the two-coil system. These can be determined for a given value of *k* by knowing the natural resonant frequency of each coil in their uncoupled (k=0) state.

For the special case of identical coils where $L_{1}=L_{2}$ and $C_{1}=C_{2}$, or more specifically when their resonant frequencies are matched $\omega_{1}=\omega_{2}$, Equation (Equation 6) simplifies to give the formula for the coupled resonant frequencies $\omega_{\pm }^{\prime }$ of(7)$$\begin{array}{cc} \omega_{\pm }^{\prime } & \approx \sqrt{\frac{1}{C_{1}L_{1}\left[1\pm k\right]}}=\omega_{1}\sqrt{\frac{1}{1\pm k}}. \end{array}$$

Equation (Equation 7) can be used to predict the change in the resonant frequencies as a function of *k* [25]. Figure 1b shows how the magnetic coupling *k* affects the resonant frequency and how it produces the resonant frequency-splitting phenomenon that separates the resonant peaks, shown as white dashed lines in Figure 1c, as a function of the coupling coefficient. The system can be thought of as exhibiting independent vibrational modes. Figure 1c also shows a red dotted line representing the dispersion separation threshold—the coupling coefficient of this system—above which resonant peaks can be resolved as distinct peaks. This threshold is dependent on the quality factor of the systems and, as such, lower resistance systems exhibit sharper, more easily resolvable curves at lower coupling coefficients. There is also a practical upper threshold to the coupling coefficient of a realistic inductively coupled system which is dependent on the geometry of the system and the permeability of the cores used within the coils.

From Equation (Equation 7), an expression for the coupling coefficient, *k*, between identical coils can be derived as a function of the measurable resonant frequencies observed in the spectra:(8)$$\begin{array}{cc} k & \approx \frac{f_{-}^{2}-f_{+}^{2}}{f_{-}^{2}+f_{+}^{2}}\;\approx \frac{\gamma_{\pm }^{2}-1}{\gamma_{\pm }^{2}+1} \end{array}$$
where $\gamma_{\pm }=f_{-}/f_{+}$ is the bimodal resonant frequency ratio. Equation (Equation 8) matches the generalised Cohn–Matthaei formula for coupled resonators derived by Tyurnev 2007 [37]. Comparable formulas are regularly applied in the modelling of microwave bandpass networks [38] and meta-material design [39], but the features and properties of coupled resonators are yet to be explored in inductive measurement applications. Rearranging Equation (Equation 7) gives an expression for the bimodal frequency ratio, $\gamma_{\pm }$, between split resonant frequencies, as a function of *k*, for an identical coil configuration:(9)$$\gamma_{\pm }=\frac{f_{-}}{f_{+}}\approx \sqrt{\frac{1+k}{1-k}}.$$

### 2.3. Coupling Coefficients

Using the centre of magnetism (CoMag) approach detailed in [40], a 2D approximation for the coupling coefficient, *k*, between neighbouring coils (see Figure 2) can be expressed as(10)$$\begin{array}{cc} k=\frac{\Phi_{2}}{\Phi_{1}} & \approx \frac{1}{4\pi }\int_{r_{2a}^{\prime }}^{r_{2b}^{\prime }}\frac{1}{r^{\prime }}\;dr^{\prime }\approx \frac{1}{4\pi }\ln\left[\frac{r_{2b}^{\prime }}{r_{2a}^{\prime }}\right], \end{array}$$
where $\Phi_{n}$ is the magnetic flux through coil *n*, $r_{1}$ and $r_{2}$ are the primary (driver) and secondary (passive) coil radii, *h* is coil height, *a* is coil separation distance, and θ∈ [π/2,π] (θ=π−ϕ) is the relative angle. Variables $r_{2a}^{\prime }$ and $r_{2b}^{\prime }$ are the distances from the CoMag to the centre of the cross-section of each side of the coil (subscript *a* and *b* denoting the near and far sides of the passive coil, respectively), and can be defined as(11)$$\begin{array}{cc} r_{2a}^{\prime } & =r_{2}\Lambda \sqrt{2\left(1-\cos\theta \right)}, \end{array}$$(12)$$\begin{array}{cc} r_{2b}^{\prime } & =r_{2}\sqrt{2\left[2+2\Lambda \left(1-\cos\theta \right)+\Lambda^{2}\left(1-\cos\theta \right)\right]}, \end{array}$$
where(13)$$\Lambda =\zeta +\eta \tan\frac{\phi }{2},$$
is an intermediate term relating to the geometric ratios of coil configuration, with $\zeta =a/r_{2}$ as the separation ratio, and $\eta =h/r_{2}$ is the aspect ratio of the coil. Substituting the above expressions into Equation (Equation 10) allows us to calculate *k* as a function of the separation, *a*, or relative angle θ. Three specific cases can be considered: co-planar separation (θ=π), the angular displacement of planar coils ($h\ll r_{2}$), and angular displacement of solenoid coils ($h\gg r_{2}$).

#### 2.3.1. Co-Planar Separation

When identical coils are co-planar (θ=π), Λ=ζ, and Equations (Equation 11) and (Equation 12) can be simplified to $r_{2a}^{\prime }=2a$ and $r_{2b}^{\prime }=2r_{2}\left(\zeta +1\right)$. We can therefore define the coupling coefficient from Equation (Equation 10) as a function of the dimensionless separation ratio, ζ:(14)$$k\approx \frac{1}{4\pi }\ln\left(\frac{\zeta +1}{\zeta }\right)\equiv \frac{1}{4\pi }\ln\left(1+\frac{r_{2}}{a}\right).$$

Equation (Equation 14) is valid for all coil aspect ratios. Equation (Equation 14) can be rearranged to arrive at a generalised linear function of the form $y=p_{1}x+p_{2}$:(15)$$e^{4\pi k}\approx p_{1}\frac{1}{a}+p_{2},$$
where $p_{1}$ and $p_{2}$ are unknown coefficients of the first-order polynomial and can be fitted to experimental data to enable displacement prediction [40].

#### 2.3.2. Angular Displacement—Planar Coils

Consider two planar coils with a varying angle as is shown in Figure 2, with $h\ll r_{2}$, such that the coil aspect ratio can be considered negligible (η≪ζ) and the coils are planar. In this case, for an angle ϕ away from co-planar existence (i.e., θ<π), Λ≈ζ, such that Equations (Equation 11) and (Equation 12) can be simplified to express the coupling coefficient, as follows [40]:(16)$$\begin{array}{c} k\approx \frac{1}{8\pi }\ln\left[1+\frac{2}{\zeta }+\frac{4}{\zeta^{2}\left(4-\phi^{2}\right)}\right], \end{array}$$
where ϕ is given in radians. Equation (Equation 16) can be simplified into a generalised linear function for the angular displacement, ϕ, for filament coils:(17)$$\begin{array}{c} e^{8\pi k}\approx p_{1}\frac{1}{\left(4-\phi^{2}\right)}+p_{2}, \end{array}$$
where again $p_{1}$ and $p_{2}$ are the unknown coefficients of a first-order polynomial, which can be found by fitting to experimental data [40].

#### 2.3.3. Angular Displacement—Solenoid Coils

For the case when a≪h and ζ≪η, a first-order approximation can be derived for *k* as [40](18)$$k\approx \frac{1}{4\pi }\ln\left[1+\frac{2}{\eta \phi }\right].$$As before, a generalised linear function for ϕ can be defined as(19)$$\eta e^{4\pi k}\approx p_{1}\frac{1}{\phi }+p_{2}.$$

Equations (Equation 14) and (Equation 18) provide simple functions with which to predict ϕ between coils when *k* is experimentally measurable. While the extreme approximations used to arrive at these functions mean that they may not provide accurate absolute values, their generalised functions given in Equations (Equation 17) and (Equation 19) can be used to fit experimental data [40].

## 3. Materials and Methods

Finite element modelling (FEM) and experimental tests were conducted to validate and characterise the bimodal resonance behaviour predicted by the circuit theory (Section 2.2). The complete details of the modelling and experimental methods are given in the following subsections.

### 3.1. Sensor Designs

To test the previously defined equations, two different extremes of coils were developed, with two different studies: co-planar separation and angular displacement.

For construction simplicity, the solenoid coil was designed with a single layer of forty turns with a wire diameter of 0.56 mm, and the planar coil was a double-layer printed circuit board (PCB) with a rectangular shape (see Figure 3d). The electrical and geometrical parameters of the solenoid and planar coils are shown in Table 1. Experimental coil dimensions are given with an error of ±0.5 mm, and experimental circuit parameters are given with tolerances of ±5%.

The coils are connected to a resonant tank circuit [41], commonly used for radio frequency and also in signal filtering applications. The solenoid coils were fabricated in-house, while the planar coils were manufactured by PCBway, China.

The inductance of the coil was obtained using an impedance analyser TREWMAC TE3001 ( TrewMac Systems, Adelaida, Australia). For the 2D model and 3D model, the geometry coil analysis module in COMSOL 6.1 was used to obtain the values of the inductances, and the capacitor and resistor used the experimental values obtained previously.

### 3.2. Finite Element Modelling

Parametrized finite element models (FEMs) were created in COMSOL 6.1 using the AC/DC module. While this paper is not focused on the optimization of the magnetic field in inductors, the magnetic coupling of the inductors is directly related to the generation of the bimodal phenomenon. Models were developed using magnetic field and electrical circuit studies to simulate the bimodal phenomenon. These are compared to experimental measurements of two extreme coil geometries, planar and solenoid coils.

The parameters used for the modelled bimodal sensors are shown in Table 1. Figure 4 shows an example simulated frequency spectrum and how the magnetic flux density, *B*, changes within an over-coupled two-coil system as a function of frequency, *f*. It is evident that at the first resonant peak ($f=f_{+}$), the magnetic flux is predominantly emitted from the driver coil (Figure 4ii), while when $f=f_{-}$, the flux is emitted predominantly from the passive coil (Figure 4iii).

The magnetic flux density shown in Figure 4ii is maximum at the first resonance peak, which corresponds to a maximum *B*-field in both the driver and passive coil, generating maximum power transfer. This effect of maximum power transfer has already been explored by Zhang et al. [42].

Figure 4i shows how the *B*-field is concentrated around the driver coil (left) for $f<f_{+}$ of the coils. Figure 4ii shows that at $f=f_{+}$, *B* increases significantly for both coils but the spatial distribution remains comparable. Figure 4iii shows that *B* is concentrated in the passive coil at $f=f_{-}$, and Figure 4iv shows how *B* is distributed evenly at higher frequencies beyond the resonant peaks.

### 3.3. Experimental Measurements

The two sensor configurations defined in Table 1 had their bimodal impedance spectra evaluated experimentally as a function of their co-planar separation, *a*, and angular displacement, ϕ. The equipment and methods used are summarised below.

#### 3.3.1. Data Acquisition

A TREWMAC TE3001 (TrewMac Systems, Australia) impedance analyser was used to measure the impedance across the driver coil. The impedance analyser took a total of 1024 points in a frequency sweep between 0.3 and 0.5 MHz for the solenoid coils and between 1.7 and 2.2 MHz for the planar coils. These data were recorded, and, using an automated peak detection process, the frequencies of the bimodal resonant peals were determined. For each angle, a frequency sweep is recorded using the impedance analyser, and the relative differences between the bimodal resonance peaks are determined. As shown in Figure 3b, the coils are contained in a plastic fixture of a wall thickness of 1.5 mm which generates an initial separation between the coils. The results of the initial impedance frequencies for solenoid and planar coils are shown in Figure 5.

#### 3.3.2. Co-Planar Separation, *a*

An X-LSM200A-E03 (Zaber, Vancouver Canada) linear stage was attached to the passive sensor (see Figure 3b), and moved in increments of 10 mm, within a coil centre-to-centre distance range of 15–75 mm (beyond which the coupling is too low for the resonance frequency to be observed in the resonance frequency sweep). A second measurement study was conducted with separation increments of 1 mm to provide a higher-resolution dataset used to predict the separation between the sensors.

#### 3.3.3. Angular Displacement, ϕ

An X-RSB120AU rotational stage (Zaber, Vancouver, BC, Canada) was used to incrementally rotate a passive resonator coil relative to a fixed driver coil, with the centre of rotation at the vertex of the shared corner, as shown in Figure 3c for solenoid coils and in Figure 3e for planar coils. Two sets of measurements were conducted: Firstly ϕ, was varied between 0 and 70° in 10° increments; the resonant frequencies were recorded and a best-fit function (derived from the formula in Section 2.3) was applied to the data. The resonant frequencies were measured again, this time for ϕ in increments of 1°, and the best-fit functions (see Section 2.3) were used to invert ϕ.

## 4. Results and Discussion

Experimental and FE results are compared directly for each of the studies, exploring co-planar separation, *a*, and angular displacement, ϕ. The results show findings varying the co-planar separation, *a*, for solenoid coils, and ϕ for both solenoid and planar coil configurations. The results are compared to best-fit functions defined in Section 2.

The experimental impedance spectra of the bimodal sensors for the two coil geometries are compared to the FE models in Figure 5. Based on the resonant splitting phenomena, the relationship between bimodal resonant frequencies and displacements *a* and ϕ can be evaluated.

Equation (Equation 8) is used to calculate *k* from measurable resonant frequencies, $f_{-}$ and $f_{+}$, and then the first-order approximations (Section 2.3) can be used to estimate displacement variables.

### 4.1. FE Model Validation

The experimental impedance spectrum results were compared to FE simulations in 2D and 3D, showing differences in the values of capacitance, as shown in Table 1, due to the separation of the wire in the hand-wound coils. These changes in capacitance lead to a different resonant frequency. To match the resonant frequencies, it is necessary to introduce a calibration value of capacitance for the resonators. This allows the simulated system to resonate at the same frequencies as the experiment.

For the solenoid and planar coils, the resonant frequencies for the three different results are approximately matched in frequency value. The magnitude depends on the impedance of the system; for the solenoid coil, there is a better approximation of magnitude than for the planar coil. Figure 5a shows the frequency spectrum for the solenoid coils with an initial centre-to-centre separation of 1.5 mm; this separation *a* represents the separation between the edges of the windings in the coils (see Figure 2).

For an initial separation of a=3 mm between the solenoid windings due to the plastic case, the solenoid coils produce the initial bimodal resonance frequency spectrum shown in Figure 5a. Planar PCB coils, with an initial separation of a=2 mm, produce the bimodal frequency spectrum shown in Figure 5b. When the coil decreases in size, the number of turns reduces, making it harder to match the finite element simulation with the experimental data due to the homogeneous multiturn approximation.

The difference between experimental and simulated results is predominantly due to disparities between real and simulated values for key variables—for instance, the ferrite permeability, lumped circuit component values, and other estimations that were made in the simulation. While the spectral responses simulated do not precisely map onto the experimental values, the simulations are sufficient to demonstrate the operational principle of using the resonance splitting frequencies for displacement applications.

### 4.2. Co-Planar Separation, *a*

The bimodal resonant frequencies $f_{-}$ and $f_{+}$ were used to determine *k* from Equation (Equation 8) as a function of *a*. As *a* increases, *k* decreases, causing $f_{-}$ and $f_{+}$ to converge in the frequency spectrum until they merge, as demonstrated in Figure 1b.

For the 2D and 3D FE, the bimodal frequency spectrum merges into a single frequency at separations greater than 12.5 mm and 17.5 mm, respectively, where the passive coil has minimal interaction with the driver coil. This phenomenon has been explored by Zhang et al. [21] for wireless power transfer applications.

Despite the differences in the impedance frequency spectrum for the 3D and 2D simulation (Figure 5), the resonant frequencies exhibit similar trends as a function of changes in coupling, as shown in Figure 6c.

### 4.3. Angular Displacement, ϕ

The results in Figure 7a(ii) show how the resonant frequencies $f_{+}$ and $f_{-}$ get closer together as a function of ϕ for the solenoid coils, showing comparable results to the evaluation of co-planar separation, *a*. In this case, *k* decreases as ϕ increases. Figure 7b shows $f_{+}$ and $f_{-}$, as a function of ϕ for the planar coils. In this case, *k* increases as ϕ increases.

The *k* between the solenoid coils varies between 0.15 and 0.35, as shown in Figure 7a(iii). This is relatively high for non-coaxial coils, where a typical value is k<0.1 [43]. These high *k* values are due to the large ferrite cores and the significant magnetic bridging it causes between the coils. The planar coils exhibit more modest values of *k*, between 0.9 and 0.14. As *k* between the planar sensors is relatively low compared to the solenoid coils (see Figure 7b(iii)), the matching between the FE models and experiment becomes more challenging, leading to an offset between results for *k* of the planar coil geometry.

The angular separation of the ferrite cores in solenoid coils increases the distance between the centres of magnetism of the cores (see Section 2.3), decreasing *k*. Conversely, the distance between the centres of magnetism for the planar coils reduces as ϕ increases, causing an increase in *k*. These trends are clearly shown in Figure 7a(iii),b(iii). Figure 7 therefore shows excellent agreement between the experimental and FE simulations, with $f_{+}$ and $f_{-}$ changing as predicted in Section 2.3.

### 4.4. Displacement Prediction

The experimental displacement measurements shown in Figure 6 and Figure 7 for co-planar separations, *a*, and angular displacements, ϕ, with increments of Δa=10 mm and Δϕ=10°, respectively, were used to fit the first-order functions defined in Section 2.3.

From the values of *k* calculated using Equation (Equation 8), linear functions of the form $y=p_{1}x+p_{2}$ were plotted based on Equations (Equation 15), (Equation 17) and (Equation 19), as defined in Table 2 and [40]. Linear regression was then used to calculate the coefficients $p_{1}$ and $p_{2}$ shown in Table 2. Note that the factor of 4 in Equation (Equation 15) was omitted as this gave rise to a stronger linear correlation as found by [40]. This disparity is likely due to the simplicity of the assumptions employed in the derivations of the first-order functions derived in [40] (see Figure 8).

The best-fit functions defined by Table 2 were rearranged to produce the best-fit curves (green dashed) for *k* as a function of the displacement variables as shown in Figure 6c and Figure 7iii. The best-fit curves based on these first-order approximate functions exhibit excellent agreement with the experimental data over a wide range of displacements in all cases. The only exception occurs when ϕ<10° in the solenoid coils, where the assumptions and simplifications of the first-order model break down. Studies were conducted to evaluate the effectiveness of using these functions to predict the physical displacement between sensors based on the experimental measurements of the bimodal resonant frequencies ($f_{-}$, $f_{+}$).

#### 4.4.1. Evaluating Separation, *a*

Experimental measurements, made every Δa=1 mm, were used, and $f_{+}$ and $f_{-}$ were recorded and used to calculate the *a* using Equation (Equation 8) and the best-fit function defined in Table 2. The prediction of *a* has a limit of a=25 mm, beyond which *k* becomes negligible and $f_{+}$ and $f_{-}$ are no longer detectable, as demonstrated in Figure 1b,c.

#### 4.4.2. Evaluating Angle, ϕ

The $f_{+}$ and $f_{-}$ peaks were recorded for variations in ϕ in both solenoid and planar coil sensors every Δϕ=1°, and the best-fit functions (Table 2) were used to predict ϕ; the results are shown in Figure 9a.

#### 4.4.3. Prediction Error, |σ|

Figure 8a compares the predicted coil separation, *a*, using the fitted function in Table 2 compared to that recorded experimentally from the linear translation stage. The absolute error, |σ|, between the known and predicted values of *a* is shown in Figure 8b. The results show that the measurement is accurate to within ±1 mm over the full range of distances measured.

For both solenoid and planar coils, the predictions are more accurate for ϕ>10°. At low angles (ϕ<10°), the planar sensors do not experience a significant change in *k* (see Figure 7b(iii)). Hence, the error in the prediction of ϕ is larger for small angles. Conversely, *k* changes significantly at low angles for the solenoid coils; however, the function employed to fit the solenoid angular displacement has low accuracy at low angles and high angles due to the simplicity of the assumptions employed (see [40]). Additionally, the solenoid coil error increases at angles ϕ>55°. This is due to the low gradient of change in the coupling coefficient *k* preventing accurate interpolation of inverted angles. At these low gradients of change in *k*, environmental factors are likely to play an important role in the accuracy of angle inversion. These may include thermal fluctuations and background RF interference. However, the most significant source of noise will likely be due to the sensors’ proximity to metallic structures, which can cause the resonant frequency of either or both sensors to shift, potentially resulting in a false-positive angular measurement. As shown in Figure 9b, the absolute error in angle for planar sensors decreases to less than ±1° for angles ϕ>10°. Despite the coil’s unoptimised design, the planar coils exhibit excellent prediction capability between 10° and 70°.

## 5. Conclusions

Pairs of over-coupled inductive coils were used to produce bimodal resonance (frequency-splitting) spectra and resonant frequency tracking was used to evaluate the separation and angular displacement between the coils. The study demonstrated the displacement-dependent bimodal phenomenon in two coil designs experimentally and highlighted the validity of a “centre-of-magnetism” method for defining first-order analytical functions of the coupling interactions between coils; 2D and 3D FE simulations were evaluated, showing excellent agreement with experimental results, and direct inversion of displacement variables showed the technique was able to resolve separations and angles to within ±1 mm and ±1°.

While these resolutions are an order of magnitude larger than Hall-effect techniques, the proof of principle in these unoptimised sensor designs demonstrates significant potential for further development of this novel sensing approach. The bimodal resonance tracking method proposed has the potential to be made in real time, deployed directly onto flexible substrates, and expanded to higher degrees of freedom, making it valuable in applications such as robotics, human–computer interaction, and non-destructive testing.

## 6. Patents

The work presented relates to a pending UK patent application—Inductive Coil Array—UK patent application no. 2215187.2, filed on 14 October 2022.

## Figures and Tables

**Figure 1 sensors-25-01822-f001:**
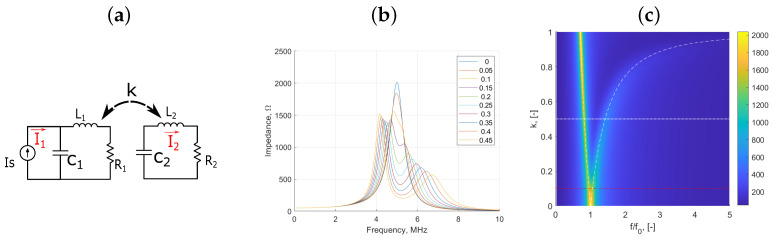
Equivalent circuit modelled results of a two-coil over-coupled inductor system, showing (**a**) the equivalent circuit diagram, (**b**) the circuit model’s predicted bimodal resonance phenomenon exhibited in the electrical impedance magnitude spectra as a function of the coupling coefficient, *k*, (from Equation (Equation 8)), and (**c**) impedance magnitude heat map showing the trajectories of resonant peaks as a function of *k*. Circuit component values used are $R_{1}=R_{2}=100\;\Omega$, $C_{1}=C_{2}=1\;\mathrm{nF}$, and inductance of $L_{1}=L_{2}=160\;\mu H$.

**Figure 2 sensors-25-01822-f002:**
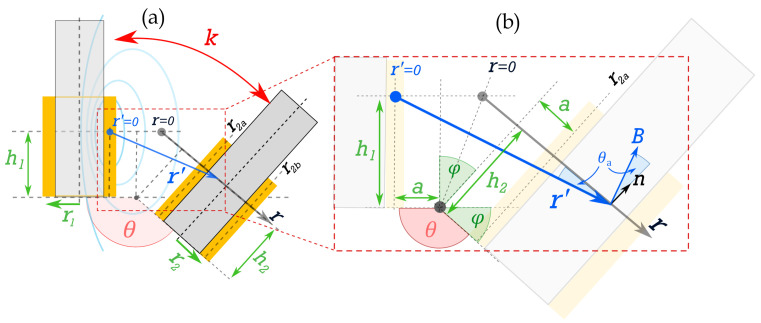
Diagram of coupled 2D ferrite-cored solenoid coils, where the yellow blocks represent the cross-section of the coil, and grey cores represent ferrite: (**a**) angular separation θ between the centre of magnetism, *h*, of each coil, and (**b**) a zoomed-in diagram of the trigonometric relationship between the distance from the line of symmetry (r=0 mm) and the distance from the centre of the B-field source ($r^{\prime }=0\;\mathrm{mm}$), showing separation, *a*, from the centre-point (r=0 mm), and angular displacement ϕ=π−θ (from [40]).

**Figure 3 sensors-25-01822-f003:**
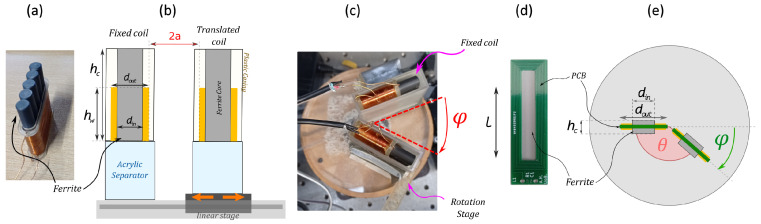
Coil design and experimental setup for co-planar separation and angular displacement studies. (**a**) Solenoid ferrite-cored coil top-down design, along with (**b**) cross-sectional co-planar separation (*a*) and (**c**) angular displacement (ϕ) testing configurations. (**d**) PCB-type (planar) coil top-down design and (**e**) angular displacement experimental configuration.

**Figure 4 sensors-25-01822-f004:**
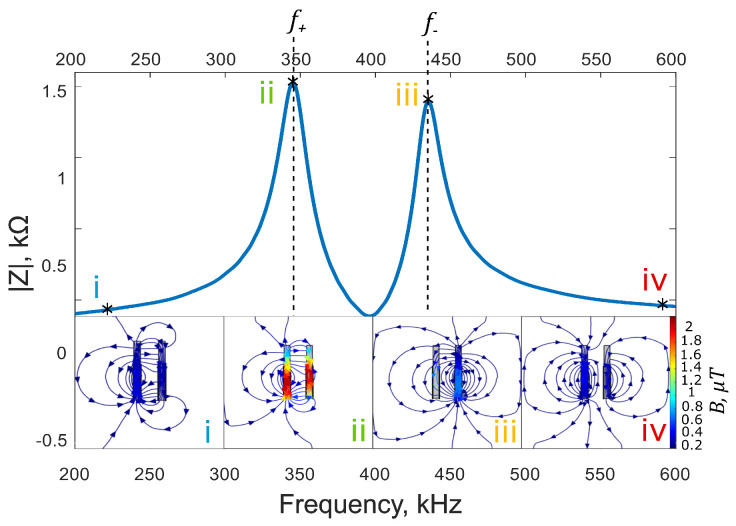
Finite element simulation of the magnetic flux density, *B*, in a bimodal resonant system in relation to the electrical impedance magnitude |Z| as a function of frequency, *f*. The spatial distribution of the in-plane *B*-field is shown at (i) 225 kHz, (ii) 350 kHz (first resonant peak, $f_{+}$), (iii) 450 kHz (second resonant peak, $f_{-}$), and (iv) 590 kHz.

**Figure 5 sensors-25-01822-f005:**
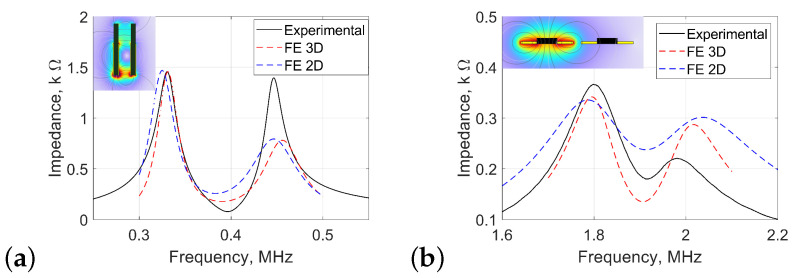
Comparison between simulated and experimental electrical resonant impedance spectra in an over-coupled driver coil. Graphs compare experimental results to 2D and 3D finite element modelled results (blue and red dashed curves, respectively) for (**a**) solenoid coils and (**b**) planar coils.

**Figure 6 sensors-25-01822-f006:**
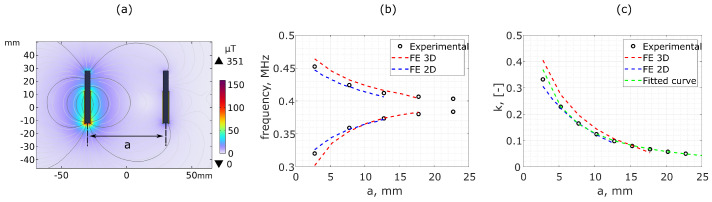
Bimodal resonant frequencies as a function of co-planar separation, *a*: (**a**) 2D finite element modelled magnetic flux density and field lines between bimodal sensor elements for a separation of a=60 mm in solenoid coils. (**b**) Bimodal resonant frequencies, $f_{-}$ (lower curves) and $f_{+}$ (upper curves), of the system as a function of angular displacement, comparing experimental (black circles) to 2D (blue dashed) and 3D (red dashed) finite element simulated results. (**c**) The coupling coefficient, *k*, calculated using measurements of $f_{-}$ and $f_{+}$ from Equation (Equation 8), where the green dashed line represents the experimental best-fit function, as defined by Equation (Equation 15) in Section 2.3.

**Figure 7 sensors-25-01822-f007:**
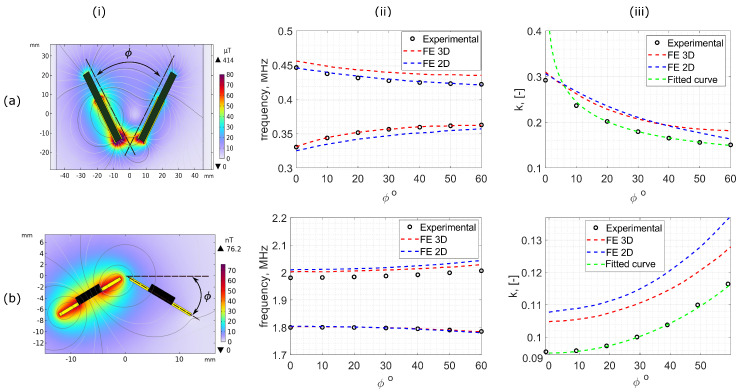
Bimodal resonant frequencies as a function of angular displacement, ϕ, in (**a**) solenoid and (**b**) planar coils: (**i**) 2D finite element modelled magnetic flux density and field lines between bimodal sensor elements for a separation of ϕ=60° in solenoid coils. (**ii**) Bimodal resonant frequencies, $f_{+}$ (lower curves) and $f_{-}$ (upper curves), of the system as a function of angular displacement, comparing experimental (black circles) to 2D (blue dashed) and 3D (red dashed) finite element simulated results. (**iii**) The coupling coefficient, *k*, calculated using measurements of $f_{-}$ and $f_{+}$ from Equation (Equation 8), where the green dashed line represents the experimental best-fit function, as defined by Equations (Equation 19) and (Equation 17) in Section 2.3.

**Figure 8 sensors-25-01822-f008:**
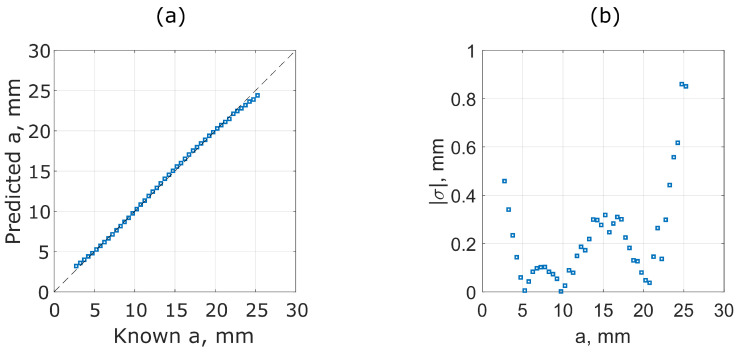
Displacement predictions of bimodal sensors, for co-planar separation, *a*, between solenoid coils. (**a**) The predicted displacement (calculated from experimental measurements and using the fitted functions in Table 2) against the known displacement values. Black dashed lines represent ideal prediction and (**b**) shows the accuracy of the predictions by plotting the absolute error |σ| between predicted and known values as a function of linear displacement.

**Figure 9 sensors-25-01822-f009:**
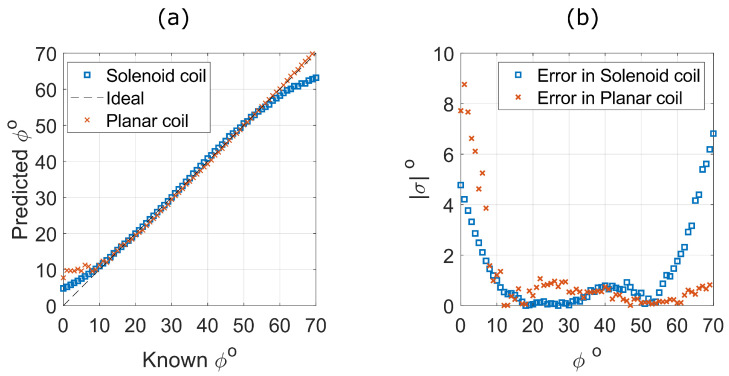
Displacement predictions of bimodal sensors with angular displacement, ϕ, between PCB and solenoid coils. (**a**) shows the predicted displacement (calculated from experimental measurements and using the angular fitted functions in Table 2) against the known displacement values. (**b**) shows the accuracy of the predictions by plotting the absolute error |σ| between predicted and known values as a function of angular displacement.

**Table 1 sensors-25-01822-t001:** Experimental (Exp.) and simulated sensor coil dimensions and electrical circuit parameters for solenoid (Sol.) and planar (Plan.) coils in experimental tests, as well as 2D and 3D finite element simulations.

Coil Dimensions (mm)					
Coil Type	*l*	$h_{c}$	$h_{w}$	$d_{in}$	$d_{out}$
Sol	43	40	25	8	11
Plan.	20	1.5	1.0	5	12
**Circuit Parameters**					
Coil Type	Turns	*L* (μH)	*C* (nF)	*R* (kΩ)	*I* (mA)
Sol. (2D)	40	708	0.2	0.1	10
Sol. (3D)	40	170	1	1	10
Sol. (exp.)	40	166	1	1	10
Plan. (2D)	28	7.9	0.9	0.01	1
Plan. (3D)	28	7.5	1	0.01	1
Plan. (exp.)	28	6.7	1	0.01	1

**Table 2 sensors-25-01822-t002:** Linear regression of the function $y=p_{1}x+p_{2}$ for the displacement of the coils as shown in Figure 6 and Figure 7.

Displacement	*y*	*x*	$p_{1}$	$p_{2}$
*a*	$e^{\pi k}$	1/a	0.39	−0.025
ϕ (sol.)	$e^{8\pi k}$	1/ϕ	3.98	3.75
ϕ (plan.)	$e^{4\pi k}$	$1/(4-\phi^{2})$	81.83	−9.40

## Data Availability

Data are available at the University of Bristol data repository, data.bris, at https://doi.org/10.5523/bris.nlizgito6ber2vnk8tkyhfl98. Accessed on 9 March 2025.

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
