# Peer review of "Displacement Sensing Using Bimodal Resonance in Over-Coupled Inductors"

_sensors, 2025, doi:10.3390/s25061822_

Round 1

Reviewer 1 Report

Comments and Suggestions for Authors

This paper mainly completes the research work on the bimodal resonance phenomenon in over-coupled inductive sensors. Through theoretical derivation, the relevant circuit models and coupling coefficient calculation methods are established. Solenoid and planar coils are designed and fabricated for experiments, and the finite element simulation and experimental results are compared and analyzed to verify the feasibility and accuracy of this technology in evaluating coil displacement.

1. The development of some key technologies in the research background is not combed deeply enough. For example, in the evolution process of inductive sensor displacement measurement technology, the main breakthroughs and deficiencies of previous studies are not described in detail.

2. In the calculation formula of the coupling coefficient, the physical meanings of some intermediate variables can be explained more deeply.

3.  In the experimental part, there is still room for improvement in the diversity of samples. Coil samples with different sizes, turns, and wire diameters, as well as magnetic cores of different materials can be considered to more comprehensively verify the applicability of the theoretical model in various situations.

4.  Explore the influence mechanism of environmental factors on the bimodal resonance phenomenon and coupling coefficient of inductive sensors, evaluate the error range that these factors may introduce, and discuss whether it is possible to establish a corresponding compensation model.

5. A certain proportion of recent research results need to be added in the references.

Reviewer 2 Report

Comments and Suggestions for Authors

This paper presents an analytical approach for estimating coil separation in an accurate way. The theoretical developments are convincing and clearly explained. The citations are relevant to the topic. I particularly appreciated the validation carried out by comparing simulation results to experimental data. The discussion of the results obtained is also well conducted. I propose to accept this very good contribution as it is.

Reviewer 3 Report

Comments and Suggestions for Authors

The authors of this paper have proposed a very practical theory and experimental study of the frequency splitting phenomena generation in mutually over-coupled inductive sensors. This work perfectly fits the scope of the journal, which is reflected in such Sensors publications as doi.org/10.3390/s16081229 or doi.org/10.3390/s20041066 (not mentioned in this article). I would advise implementing some minor revisions prior to the publication of this work.
a) According to the provided graphs, it is evident that the proposed method exhibits low errors not only within the lower limit of the angle range but also at the upper limit. The authors have not explicitly addressed this observation in either the main text or the conclusions, which could lead to ambiguities in the interpretation of the results.
b) A more detailed explanation or analysis of deviation of the theoretically calculated Z value from the experimentally measured (Figure 5) discrepancies would enhance the study.
c) Table 2 and Figure 8 are not formatted correctly and should be modified.

Round 2

Reviewer 1 Report

Comments and Suggestions for Authors

This article focuses on the research of displacement sensing by using the bimodal resonance phenomenon in over-coupled inductors. It expounds on the relevant theories, experimental procedures, and results, verifies the feasibility of this measurement technique, and provides new ideas for displacement measurement of inductive array sensors.

  1. Only two specific types of coils were selected for the experiment, resulting in a relatively single sample type. A single experimental sample cannot comprehensively verify the effectiveness of this displacement sensing method under various coil conditions.
  2. The article does not discuss the interactions among factors such as the self-inductance, mutual inductance, capacitance of the coils, and external interference. Therefore, it is impossible to fully reveal the complex mechanisms in the displacement sensing process.
  3. When deriving the formulas related to the coupling coefficient, the article adopted many approximations and simplifications. For example, in formula (10) for calculating the coupling coefficient, the center of magnetism method was used for two-dimensional approximation, which will affect the accurate prediction of displacement measurement.
  4. Although the research shows that this method can achieve a measurement accuracy of ±1 mm and ±1°, there is still room for improvement compared with some high-precision displacement measurement techniques.
  5. The references cited in the paper are relatively old, and they cannot fully reflect the latest research trends.
Comments on the Quality of English Language

The English could be improved to more clearly express the research.
